# Interaction Design Based on Big Data Community Home Care Service Demand Levels

**Fangyuan Jiang [1], Wan-Sok Jang [2] and Young-Hwan Pan [1,*]**

1   Department of Smart Experience Design, Kookmin University, Seoul 02707, Republic of Korea
2   College of Communication, Qingdao University of Science and Technology, Qingdao 266042, China
*   Correspondence: peterpan@kookmin.ac.kr

**Abstract:** Most of the contemporary models for meeting the majority of the needs of middle-aged and elderly people are community-based, in-home care. Therefore, this paper designs an Interaction model that can meet the need for a rich spiritual and cultural life of the elderly at home. First, the questionnaire content of the Chinese Longitudinal Healthy Longevity Survey (CLHLS) sampling method was designed based on the content of community-based home care services. Then, using the CLHLS sampling method, the survey results of the home care group were collected to form a community of big data consisting of four types of home care service needs. Finally, the Interaction book model was designed based on the hierarchy of service needs obtained from Abraham Maslow's hierarchy of needs classification method. The experimental results showed that the mean values of the target population's ratings for the presentation and interface aesthetics of the Interaction mode were 4.34 and 4.19, respectively, the mean value for improving the learning effectiveness of the home-bound population was 4.57, and the mean value for their overall satisfaction was 4.31. It proves that the Interaction model is ideal for practice and can meet the learning needs of the elderly, at-home population from different service demand levels, thus solving the problem of education for the elderly.

**Keywords:** community big data; service demand level; CLHLS sampling method; multi-indicator multi-factor model





## 1. Introduction

China's elderly population is growing extremely fast, and the degree of aging is deepening [1]. In addition, aging in China faces the overlapping phenomena of advanced aging, disablement, and empty-nesting [2–4]. The number of people with mild, moderate, and severe loss of self-care has risen exponentially. According to official statistics, by the end of 2020, the number of people with varying degrees of loss of self-care ability exceeded four million. There are more than two million people who have at least two chronic diseases in their bodies. At present, not only is the physical health of human beings getting closer to the direction of aging, but also the phenomenon of empty nesting is becoming more and more significant, and the number of such people is likely to double [5,6]. Such a development will also trigger a more serious situation, such as urban-rural inversion and aging before the rich, which will bring more serious and multifaceted challenges for aging in place [7,8]. Not every elderly person enjoys a happy old age, there will be a promising future for those who come after. However, with the increasing pressure of aging, it is impossible to solve the aging problem with families and institutions alone. Therefore, the community-based aging-in-place model is used as the main means to alleviate the pressure of aging in the future. With the high level of attention and devoted support of the state, the structure of the elderly system is becoming more and more complete, and the community-based elderly care model is gradually entering the development situation at the strategic level and has achieved phased results [9]. The widespread popularity of this model has not only broadened the research perspective of community-based home care

services but also improved the theoretical study of elderly care services [10]. By innovating the supply of community home care services in order to improve the relevance of service supply and service quality, the needs for elderly care are better met [11,12].

Current research on community-based senior care services has been conducted from many aspects, such as origin, connotation, and advantages. The literature [13] provided a framework for research embedded in a neoliberal context. The family aging model was tested through interviews with 100 older adults in five communities in Wuhan. The literature [14] introduced a conceptual model to explore the relationship between the determinants of demand for elderly services and social organization. Questionnaire-based structural equation modeling was used as a test method for synergistic developmental relationships. The results suggest that the study can provide a guiding direction for improving home care services. The literature [15] used logistic regression models to analyze data to test whether socio-demographic characteristics, physical health, loneliness scores, and other factors were associated with home care intentions among rural and urban older adults. Research has shown that there are differences in the willingness of older adults to take care of themselves at home where they live, and that targeted policies should be developed for different groups of older adults in order to solve the problem of excessive waste of social resources caused by the emerging and lagging management of the elderly service industry. The literature [16] proposed the introduction of the Slack Based Model-Data Envelopment Analysis (SBM-DEA) model to assess the performance of community-based home care service centers. We also built a systematic framework for the performance evaluation of several social organizations in Wuhan. The literature [17] identified five external coping strategies by examining the sustainability of geriatric care in Slovenia: use of formal care services, use of extended family networks, use of wider community networks, cohabitation, and family adjustment. A synthesis of national and international scholars found that the established literature is not better adapted to the modern socialist environment. The education needs of the elderly living at home are increasing, and it is urgent to improve the quality of education services for this group.

The special nature of the home care group determines the special nature of the educational services. In order to solve the main problems such as poor targeting, lack of precision, and the mismatch between the supply and demand of community home care services, the questionnaire content of the CLHLS sampling method was designed based on the content of community home care service. Instructional model design is an important part of achieving educational services. Using the CLHLS sampling method, the survey results of the home care group were collected to form community big data and used as the data basis for the hierarchy of senior care service needs. The structural model and measurement model are fused to build a multi-indicator, multi-factor model to analyze the influencing factors of service demand. Abraham Maslow's hierarchy of needs and the demand for community home care services are divided into four levels: life care service demand based on physiological needs, health care service demand based on safety needs, spiritual and cultural service demand based on emotional and belonging needs, and legal aid service demand based on respect needs. Based on the traditional ADDIE curriculum development model, we designed an Interaction book model for senior education and senior groups. This study expects to broaden the perspective of exploring the needs of other home care services and to improve and optimize the home care service system. By innovating the supply of big data community home care services, we can improve the quality of services and better meet the needs of the elderly at the spiritual level.

## 2. Big Data Community Home Care Service Demand Hierarchy

### 2.1. Analysis of Demand for Home Care Services

If the sampling is done in actual, equal proportions of the target population, the sample will be concentrated in female elderly and lower-aged elderly, which is not reasonable [18]. Therefore, a multi-stage, unequal proportion of CLHLS target sampling method was used to balance the sample size and gender proportion of different age groups. The questionnaire

content of the CLHLS sampling method was designed based on the content of community-based home care services. Using the CLHLS sampling method, the results of the survey of the elderly at home group were collected to form the community big data, and they were used as the data basis for the hierarchy of elderly service needs. After sampling by CLHLS method, the formed community big data is shown in Figure 1.

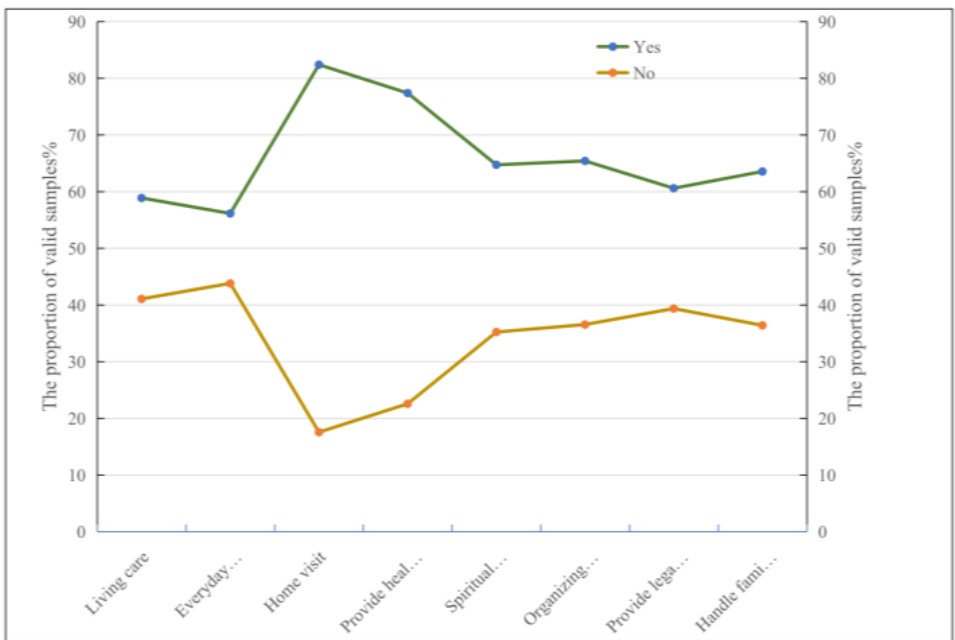

**Figure 1.** Community aging-in-place big data.

According to the community's big data, the needs of home care services can be divided into four categories: living care, medical care, spiritual comfort, and legal assistance. From Figure 1, it was illustrated that 57.84% of the sample population had living needs and 59.28% had shopping needs. The percentage of the sample population without living needs was 40.54%, and the percentage of the sample population without shopping needs was 44.03%. The percentage of people who needed home services was 83.14%, and the percentage of people who needed to learn about health care was 78.83%. The percentage of the sample who did not need home services was 16.37%, and the percentage of the sample who did not need to learn about health care was 21.25%. Regarding the demand for spiritual comfort services, which include chatting and organizing social entertainment activities, 66.54% of the sample population needed to chat to relieve boredom and 70.05% wanted to organize and participate in social entertainment activities. The percentages of those who did not need to chat to relieve boredom and did not want to organize or participate in social recreational activities were 34.67% and 35.26%, respectively. Regarding the demand for legal aid services, 77.2% and 62.31% of the sample population wanted the community to provide rights protection services and help with disputes, respectively. The percentage of those who did not want the community to provide rights protection services was 38.13%, and the percentage of those who did not need the community's help in handling disputes was 35.49%.

### 2.2. Service Requirements Hierarchy

2.2.1. Demand Influencing Factors

Based on the community big data formed by the survey results of the elderly at home group, the structural model and the measurement model are fused and a multi-indicator, multi-factor model is established to analyze the influencing factors of the service demand by analyzing the cause variables, latent variable paths, and indicator variables, in order to comprehensively measure the objective values of latent variables and accurately classify the

demand levels. Assuming that the demand for educational services is $\eta$ (i.e., latent variable) and the basic information parameters such as health status, self-care ability, and economic level of the elderly population are $x_q$ (i.e., cause variables), the equation expressions of the structural model in the model is:

$$\eta = \gamma_1 \chi_1 + \gamma_2 \chi_2 + \ldots + \gamma_q \chi_q + \xi \tag{1}$$

In the above equation, $\xi$ is the error term and is used as a compensation parameter.

Assuming that the demand for the four services is $y_p$ (i.e., the indicator variable) and the error term is $\varepsilon_P$, the equation expression for the measurement model in the model is:

$$y_1 = \lambda_1 \eta + \varepsilon_1 \tag{2}$$

$$y_p = \lambda_p \eta + \varepsilon_\rho \tag{3}$$

The measurement model reflects a linear relationship between the demand for educational services and the demand for four services, including life care. After simplification, a simplified equation of the two models is obtained. It is as follows:

$$\eta = \gamma' x' + \xi \tag{4}$$

$$y' = \lambda' \eta + \varepsilon' \tag{5}$$

where $x' = (x_1, x_2, \ldots, x_q)$ and $y' = (y_1, y_2, \ldots, y_p)$ are the sets of variablesof observable exogenous causes and endogenous indicators, and $\gamma' = (\gamma_1, \gamma_2, \ldots, \gamma_q)$ and $\lambda' = (\lambda_1, \lambda_2, \ldots, \lambda_p)$ are the sets of structural parameters of the structural and measurement models. All other parameters are error terms. The simplified two-model defining equations after the union are obtained are:

$$y = \Pi x + v \tag{6}$$

Among them.

$$v = \lambda' \xi + \varepsilon \tag{7}$$

$$\Pi = \gamma \lambda' \tag{8}$$

The measurement errors are assumed to be normally distributed and independent of each other in the above set of equations, expressed as [19–21]:

$$E = (\xi) = E(\varepsilon) = 0 \tag{9}$$

$$E = \left( \xi^2 \right) = \sigma^2 \tag{10}$$

$$E = (\varepsilon \varepsilon') = \Theta^2 \tag{11}$$

$$E = (vv') = E\left[ (\lambda \xi + \varepsilon)(\lambda \xi + \varepsilon)' \right] \sigma^2 \lambda \lambda' + \Theta^2 \tag{12}$$

The model focuses on the calculation of the function matrix through the covariance matrix. In this case, the covariance matrix consists of observable variables and the function matrix consists of parameters to be estimated. After substituting the observable variables, the overall covariance $\Sigma(\theta)$ is solved by using the following equation:

$$\Sigma = \Sigma(\theta) \tag{13}$$

The overall covariance $\Sigma(\theta)$ in the above equation is the covariance matrix consisting of the observable variables. The parameter to be estimated exists in the vector $\theta$. This vector can be solved by the structural model equation and the measurement structural model equation. It is combined with the structural model equation to obtain the value of the demand for educational services, i.e., the value of the latent variable.

In the parameter estimation of the structural model equation, there are usually two cases that require the use of different methods to accomplish. The parameter estimation methods for the different cases are described in detail as follows.

(1) If the model satisfies the condition of no correlation between error terms and equal variance, it is estimated using the least squares estimation method [22–24]. The basic idea of the least squares method is to find an estimate of the parameter $\beta$ that minimizes the modulus square of the error vector $e = y - x\beta$, that is, minimizes $\| e \|^2$ [25,26].

Denoted as.

$$Q(\beta) = \frac{(y - X\beta)'}{y - X\beta} \tag{14}$$

It is easy to obtain.

$$Q(\beta) = \frac{y'y}{\beta'x'x\beta} \tag{15}$$

Taking partial derivatives of $\beta$ and making $\frac{\partial Q(\beta)}{\partial \beta} = 0$, while through the matrix quotient formula we know that:

$$\frac{\partial y'x\beta}{\partial \beta} = x'y \tag{16}$$

$$\frac{\partial x'x\beta}{\partial \beta} = x'x\beta \tag{17}$$

Therefore.

$$X'y + X'X\beta = 0 \tag{18}$$

That is:

$$- X'y = X'X\beta \tag{19}$$

Equation (19) is called the canonical equation. Equation (19) has a unique solution when the rank of the design matrix $x$ is $\rho$. At this point, it can be found that:

$$\hat{\beta} = \frac{X'X}{X'y} \tag{20}$$

According to the extreme value of the function, $\beta$ is only a stationary point of the function $Q(\beta)$, so it is necessary to further prove that $\hat{\beta}$ can make $Q(\beta)$ reach the minimum. It is also necessary to prove that it is $\hat{\beta}$ that minimizes $Q(\beta)$. In fact, for any $\beta$.

$$Q(\beta) = \left\| \frac{y - X\hat{\beta}}{x(\hat{\beta} - \beta)} \right\| \tag{21}$$

Since ξ, satisfies the canonical Equation (19), the third term of the above equation is equal to 0 and the second term is non-negative. Thus, we have:

$$Q(\beta) \geq Q(\hat{\beta}) \tag{22}$$

So $\hat{\beta}$ can indeed minimize $Q(\beta)$. Then, prove that what minimizes $Q(\beta)$ must be $\hat{\beta}$. For the Equation (23) holds when and only when:

$$X'X(\hat{\beta} - \beta) = 0 \tag{23}$$

Equivalently,

$$X(\hat{\beta} - \beta) = 0 \tag{24}$$

So we have:

$$X'X\beta = X'y \tag{25}$$

(2) In the parameter estimation of the structural model equation, if the condition of no correlation between the error terms and equal variance is not satisfied, the generalized

least squares estimation is used. In this case, the covariance array of the error vector satisfies $\text{cov}(e) = \sigma\Sigma$, where $\Sigma$ is either positive, definite, or semi-positive definite, and $\Sigma$ is often not known exactly. It is assumed that $\Sigma$ is known and $\Sigma > 0$, i.e., $\Sigma$, is a known positive definite array. The fit function for generalized least squares estimation is given in Equation (26).

$$F_{\text{GLS}} = \frac{1}{2} * \text{tr}\left\{ \left[ \left( S - \sum(\theta) \right) W^{-1} \right]^2 \right\} \tag{26}$$

where $W^{-1}$ denotes the weighting matrix of the residual matrix. The weight matrix is selected in two ways:

$$W^{-1} = S^{-1} \tag{27}$$

$$W^{-1} = \left( \sum(\theta)^{-1} \right) \tag{28}$$

If $\beta^*$ satisfies:

$$\frac{y - x\beta^*}{\Sigma} = m \frac{y - x\beta}{\Sigma} \tag{29}$$

Then the parameter results of generalized least squares estimation $\beta^*$ are obtained.

### 2.2.2. Hierarchy of Needs

A multi-indicator, multi-factor model was used to analyze the factors influencing the service needs embedded in the community big data collected by the CLHLS method. According to the analysis results, the hierarchy of needs for community home care services is structured according to Abraham Harold Maslow's basic hierarchy of needs theory shown in Figure 2a, which is shown in Figure 2b from four aspects: physiological needs, safety needs, emotional and belonging needs, and respect needs, respectively. This elderly service demand hierarchy contains four categories and eight services, and the four categories correspond to the four demand levels one by one. The first level is the demand level of life care services based on physiological needs. The second level is the demand level of health care services based on safety needs. The third level is the level of spiritual and cultural services based on the need for emotion and belonging. The fourth level is the level of legal aid services based on the need for respect.

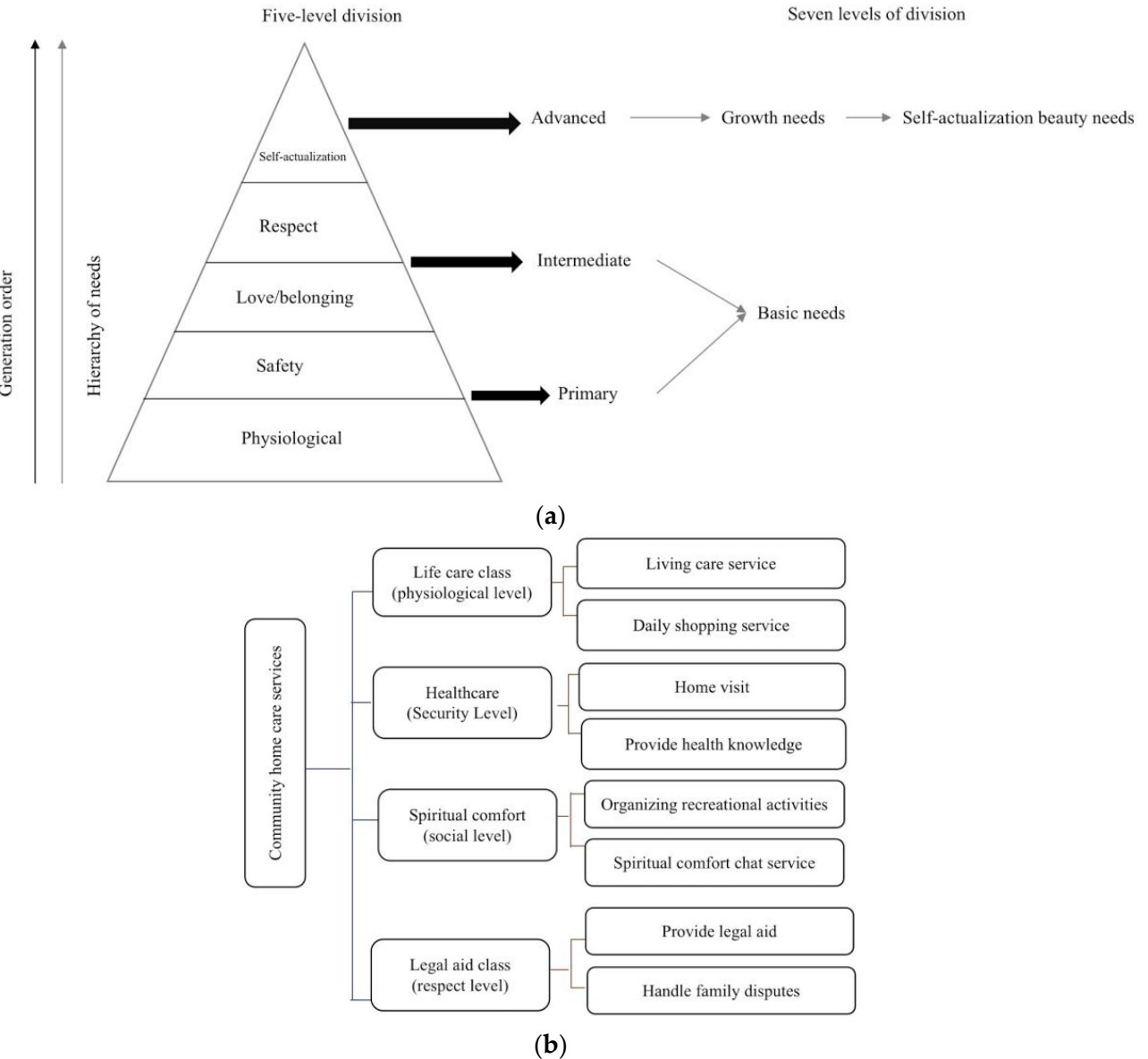

**Figure 2.** Hierarchy of demand for elderly services under the division theory. (**a**) Hierarchy of needs theory; (**b**) Hierarchy of needs for senior care services.

## 3. Interaction Book Design Based on the Hierarchy of Needs for Elderly Services

The ADDIE traditional curriculum development model of analysis, design, development, implementation, and evaluation is referenced in this study [27,28]. The education of home care groups has a certain specificity. This nature is the key basis for the design process of the Interaction book curriculum and determines the success or failure of the design outcome. It is also the main reason why this research topic is different from other Interaction book design research topics. Therefore, with reference to the ADDIE traditional curriculum development model, the Interaction book model in Figure 3 is designed for the needs of senior education services. The model focuses on analysis and design and can well meet the educational service needs of the elderly at home group.

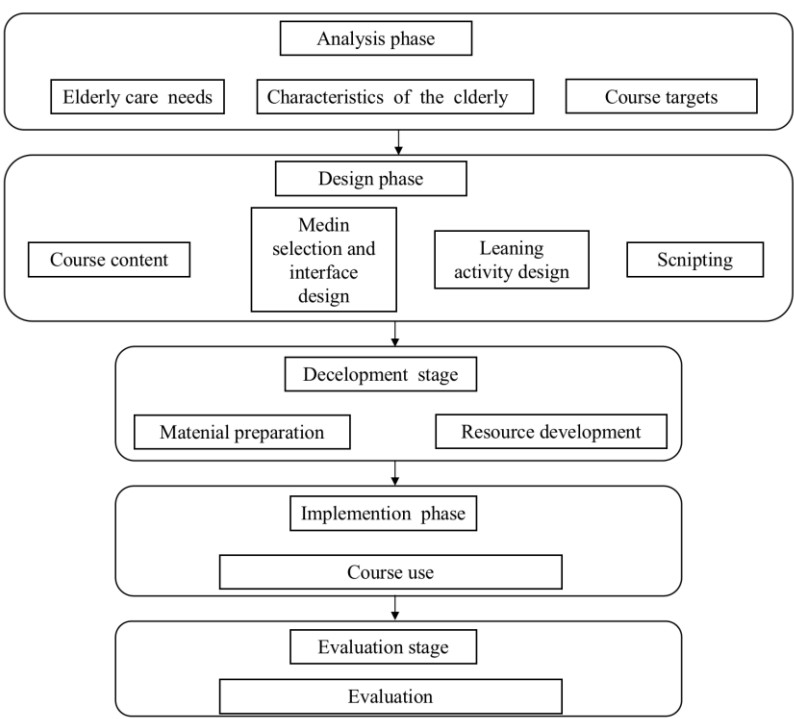

**Figure 3.** Interaction book design for the needs of senior education services.

*3.1. Analysis*

(1) The need for home care services: This stage is used to set up Interaction objectives such as content and its presentation. The pensioners do not need to face the pressure of higher education exams or employment promotions, but rather teaching and studying only to meet their needs of enjoying hobbies, a healthy life, physical and mental cultivation, and a contribution to society. For the memory characteristics of the elderly at home population, the length of the course does not exceed 15 min and the knowledge points do not exceed two. As much as possible, multimedia technology is used to provide diversified education through interactive functions.

(2) Target population characteristics: According to the characteristics of the target group, on the basis of the general design approach, incorporate the special characteristics of the group. Let the content of the Interaction book be feature-based to make it more compatible with the educational characteristics of the target group. The group has significant differences in basic cognition, learning attitudes and motivation. The influence of physical, psychological, cognitive and behavioral characteristics that have commonalities on the content of the Interaction book can be broadly summarized in Table 1.

**Table 1.** Changes in the characteristics of older adults and their distress about using online courses.

| Change Characteristics | Specific Changes | Problems Brought to the Use of Online Courses |
| --- | --- | --- |
| Physiological changes | Impaired vision | Difficult to read the Interaction content |
| | Hearing impairment | Difficult to hear the Interaction knowledge |
| | Reduced flexibility | Difficult to operate |
| Psychological changes | Increased sense of loneliness, loneliness and loss | Existing resource types and presentation forms cannot meet the requirements |
| Cognitive change | Impaired memory, intelligence, thinking ability and learning ability | Difficult to understand the content taught in a short time |

In view of the above adverse effects, targeted means can be applied to optimize the design. For example, for physiological impacts, page fonts can be enlarged, voice functions or icons can be added, color differences can be increased, and pages can be simplified. For the psychological impact, the user can diversify the teaching curriculum, add learning contents of various subjects, and combine the blended learning methods of text, graphics, sound and image, and other media to ensure the rationality and interest of teaching strategies. For cognitive impact, the user can split the knowledge points and complete the lectures one by one with clear instructions and a small amount of interaction.

(3) Interaction objectives: As the evaluation basis of teaching effectiveness, teaching objectives have four factors: learners, teaching conditions, teaching activities, and teaching standards. Its analysis perspective needs to start from the four levels of home care service needs and set up objectives based on the needs of the taught elderly group.

### 3.2. Design and Development

Design and development: The Interaction knowledge is selected for the four levels of home care service needs, and the corresponding courses that can be set up are shown in Table 2. The Interaction courses take the centrality of the elderly group as a prominent point and combine with humanistic care for the interface design [29–31]. Let the knowledge content be presented in a reasonable interface structure with multimedia forms such as text, image, audio, and animation through simple and easy-to-understand content, appropriate color matching, and suitable text and picture size. The access methods of different media forms are shown in Table 3. When the Interaction materials are in text formats such as doc, .txt, etc., the acquisition methods include mainly keyboard, scanning, voice recognition, and others. When the Interaction materials are in .jpg, .gif, and other image formats, the adopted acquisition methods are photography, screenshot, software creation, etc. When the Interaction materials are in audio, video, and animation formats such as .avi, .mov, .MP4, .gif, etc., they are mainly acquired by recording and downloading. When writing multimedia running scripts, the presentation form of Interaction contents is reasonably arranged according to the output capability characteristics of computers. In the course of the scenario-based Interaction activities, relevant resource materials that play a supportive role in the learning of older adults, such as Interaction courseware and literature, are selected [32–36]. Strategy design uses Interaction strategies of independent learning and learning strategies of organizational retelling [37–41].

**Table 2.** Relationship between service demand and Interaction content.

| Service Demand Level | Settable Interaction Content |
| --- | --- |
| Physiological level | Wellness, health care, home care, cooking, use and maintenance of electrical appliances, knitting technology, information technology, etc. |
| Security level | Social life of the elderly, geriatric psychology, psychological grooming of the elderly, second life plan, etc. |
| Social level | Piano, dance, photography, painting, calligraphy, literature, history, foreign languages, floriculture, information technology, etc. |
| Levels of respect | Relevant laws and regulations, etc. |

**Table 3.** Table of Interaction materials.

| Media Format | Format Type | Acquisition Method |
| --- | --- | --- |
| Text | Doc, htm, txt, html, etc. | Keyboard input, scanning, voice recognition, stroke writing, etc. |
| Graphical images | Gif, jpg, jpeg, png, etc. | Camera photos, screenshots, software creation, etc. |
| Audio | Mov, avi, MP3, etc. | Recording, file download, etc. |
| Video | Avi, asf, mp4, file download, etc. | Free recording, file download, etc. |
| Animation | Swf, gif, etc. | Free production, file download, etc. |

### 3.3. Implementation and Evaluation

After developing the overall curriculum of the Interaction book, the target user population was invited to use the learning course content. In order to determine whether the Interaction objectives have been achieved and the expected effect of the Interaction book design has been achieved, a comprehensive evaluation of the design level of the Interaction book curriculum is conducted based on the learning results in terms of content, structure, etc., and whether it meets the learning needs of the aging-in-place population. This is used as the basis for adjusting the curriculum-related items.

## 4. Analysis of Hierarchical Validity of Needs and Practicality of Interaction Model

### 4.1. Discussion of the Effectiveness of the Hierarchy of Needs for Elderly Services

The demand for elderly care services of the target group was analyzed using the CLHLS sampling method, and the results are shown in Figure 4. The results of the analysis of the demand for elderly care services under general conditions in this figure show that the significance value of the residence situation of the elderly at home population is 0.07. Therefore, the results of the obtained demand analysis are statistically significant. The percentage of the elderly people living with a spouse and living with children are 54.8% and 29.7%, respectively. Both of these two types of home care groups showed a relatively positive attitude toward the demand for senior care services. The correlation between their physical condition and their degree of desire for senior care service needs was significant, as evidenced by the fact that the significance value of physical condition of the target group did not exceed 0.02. The significance value of 0.001 less than 0.002 for the degree of knowledge of elderly care services indicates that although 43.6% of the people do not know anything about aging in place, their desire for services gradually increases as the degree of knowledge of elderly care services among the elderly in place population gradually increases. The significance values of economic source and education level of the elderly at home group are 0.701 and 0.457, respectively, both of which are greater than 0.07. This fully indicates that the economic source and education level of the group do not affect their desire for elderly care services.

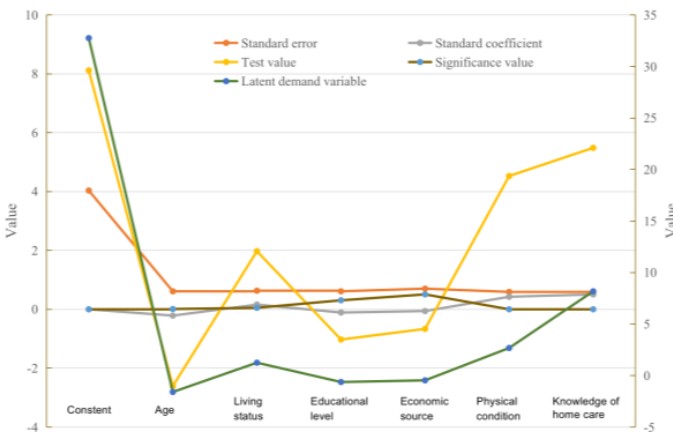

**Figure 4.** Analysis of the demand for elderly services under general conditions.

The data in Figure 5 shows that the significance values of bathing and using public vehicles for the target group do not exceed 0.07, and the significance values of shopping and medication dependence are 0.039 and 0.021 respectively, which are also under the significance level of the value of 0.07. Therefore, the results of the analysis of the correlation between the ability of daily living and the demand for elderly services are statistically significant and show a significant positive correlation. This set of data can fully illustrate that the greater the dependence of the population in bathing, using public vehicles, shopping, and taking medication, the higher their desire for senior care service needs. The results of this study also describe common characteristics of the aging-in-place population, such as

age-related slowing of physical functions. For this population, most behavioral activities are becoming a barrier to living and pose a significant safety risk. The outdoor activities are even more overwhelming for some elderly people.

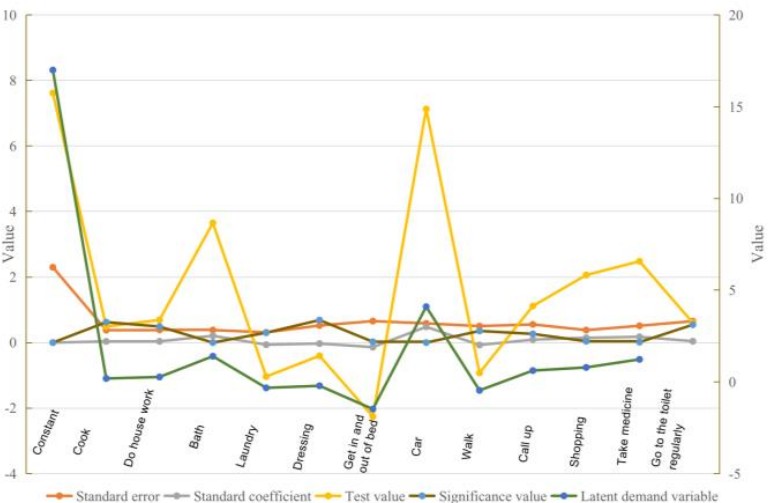

**Figure 5.** Influence of daily living ability on the demand for senior care services.

In summary, the analysis of the factors influencing the demand for community elderly care services through the multi-indicator, multi-factor model makes the classified levels of demand for elderly care services at home independent of most factors, with good immunity and validity, and can be used as a reliable theoretical basis for Interaction design.

### 4.2. Practical Discussion of the Interaction Model

In this study, an anonymous survey was used to verify the effectiveness of the application of the interaction-in-design model using the aging-in-place population as the target population. Eighty questionnaires with 100% recall and validity rate were distributed using the CLHLS sampling method to explore the feelings of the target population after using the interaction-in-book model. The survey results are shown in Figure 6.

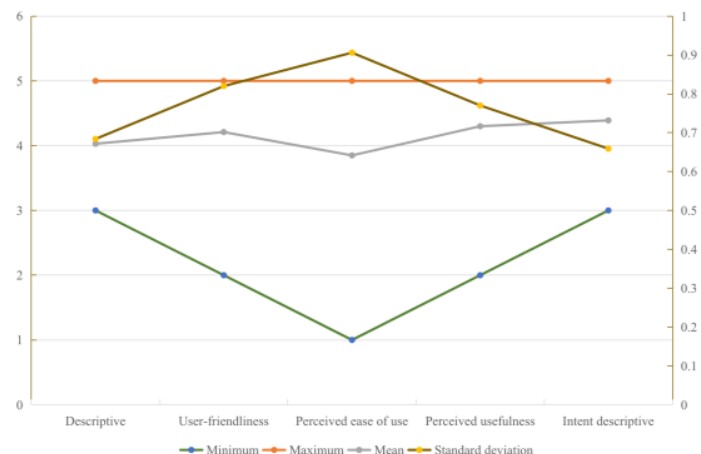

**Figure 6.** Interaction book model practicability analysis result scores.

After the statistical survey results, the following points were obtained from the target group's feelings about using the model.

(1) Visual perception: The respondents were satisfied with all the functions of the Interaction book model and gave high ratings to each feature description category so that the mean value of the questions was above 4.1. The Interaction book model is designed to meet the personalized visual preferences of the target population by adopting a

layout that approximates a paper book and designing color-matched module clusters according to the target users' favorite layout. As a result, the home-based elderly population highly rated the layout and aesthetic degree of the interface, with mean values of 4.34 and 4.19 for the two types of questions, respectively.

(2) Degree of operation: The majority of the invited survey respondents were over 60 years old, of which 26.78% were over 75 years old. This directly reduces the rating of the degree of operation by the target population. Most of the aging-in-place population has difficulties in using IT equipment due to age. Therefore, the mean rating for the degree of operation question was only 3.81. Although the mean rating for the degree of operation was lower than the mean rating for the degree of effectiveness, it was within the range of relative satisfaction.

(3) Degree of effectiveness: Regarding the effectiveness question item of the Interaction course, the aging-in-place population gave a rating of 4.1 or higher. The mean value of the question on the effectiveness of the Interaction model exceeded 4.4 because the elderly people thought that the designed Interaction model could significantly promote their interest, effectiveness and experience in learning. Survey respondents rated the Interaction mode as helpful to their learning and enabled them to achieve better learning outcomes, so they scored higher, with a mean value of 4.57. 92.5% of the elderly people at home were satisfied with this Interaction mode, with a mean satisfaction rating of 4.31. Therefore, the elderly people at home are satisfied with this Interaction model in general.

(4) Willingness to use: The statistical results of the willingness to use questions show that the mean value of each question exceeds 4.5 points. Most of the respondents were willing to participate in the development activities of the Interaction model, and they were also willing to recommend this Interaction model to others. The mean score of 4.48 for these two questions indicates that the target users are highly appreciative of this Interaction model and are willing to share it.

## 5. Conclusions

In this paper, we use CLHLS method and multi-indicator, multi-factor model to collect and analyze the demand and influencing factors of home care services. Based on the four obtained needs of home care services and Abraham Maslow's hierarchy of needs division method, the hierarchy of needs of home care services was divided. Referring to the ADDIE model, we designed an Interaction model for the elderly group. The results of the practical analysis of the hierarchical validity of needs and Interactions model showed that the sig value of housing status of the elderly is 0.07, the sig value of physical status is less than 0.02, the sig value of bathing and using public vehicles is 0.007, and the sig values of shopping and medication dependence are 0.039 and 0.021, respectively. The mean ratings of the target population on the presentation and aesthetics of the interface of the Interaction mode were 4.34 and 4.19, respectively; the mean rating on improving the learning effectiveness of the population was 4.57; and the mean rating on their overall satisfaction was 4.31. The above data show that the analysis of the factors influencing the demand for community elderly care services through the multi-indicator, multi-factor model makes the classified levels of demand for elderly care services at home independent of most factors and has good immunity and validity, which can be used as a reliable theoretical basis for the Interaction book design. The aging-in-place population is satisfied with the Interaction model as a whole and has a high willingness to share this Interaction model. This research shows that the designed teaching format is not only immune and socially useful in the current technological era, but also can contribute to the scientific research of human society by virtue of the overall satisfaction of the home-based elderly population with the teaching model, which is increasingly evolving into a significant part of the group behave or chain.

**Author Contributions:** Methodology, W.-S.J. and Y.-H.P.; Writing—original draft, F.J. All authors have read and agreed to the published version of the manuscript.

**Funding:** This work is supported by research projects of Qingdao University of Science and Technology (WST2021020) and Kookmin University.

**Institutional Review Board Statement:** Not applicable.

**Informed Consent Statement:** Not applicable.

**Data Availability Statement:** By utilizing longitudinal panel data approved in 2020.

**Acknowledgments:** This work was supported by WST2021020.

**Conflicts of Interest:** The authors declare no conflict of interest.

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
