# Peer review of "Interaction Design Based on Big Data Community Home Care Service Demand Levels"

_applsci, doi:10.3390/app13020848_

Round 1

Reviewer 1 Report

Overall, the paper is an excellent read. The logic is clearly presented and flowing from each sentence and each section.  What needs to be updated before publication are the figures.  Each of the figures have either some spacing issue in the caption (lack of space in Figure 1.) or the Figure caption is shifted to the left or right for some reason.  I would recommend black and white colours for figures and not colour. Also, the x and y axis needs better formatting. The text of the x-axis should not be in the graph as for Figures 4-5. Also, the Figures 2-3 need to be larger and in black and white to read for clarify. The numbered references in the Reference section need to be better formatted, please review the journals published articles in the appropriate formatting and alignment.  I don't have any cop/edits or major changed to the text, as the paper was well-organized, easy-to-read, and good sentence structure.  Equations valid and good formatting, but the figures and references formatting is a major issue and requires update.

Author Response

I am very grateful to you for your valuable opinions. We have revised the questions you raised in the paper.

1.The spacing in the title has been modified.

2.The color of pictures 2-3 has been modified to black and white.

3.The format of references has been modified.

Reviewer 2 Report

1. It would be a good idea to separate the Introduction and Related work sections

2. Please, could you include the main problem that you solved in the Introduction section?

3. My opinion is that your contribution must be stated in the Introduction section.

4. Please, enter the full name for the SBM-DEA model and the CLHLS sampling method.

5. It would be good to split the subsection Design, development, implementation, and evaluation into two sections (e.g., first Design, development and implementation, and second Evolution) so readers can easily follow the text.

6. Typographical errors: Figure 6 .Interaction book model practicability analysis result scores

7. Figure 3. Interaction book design for the needs of senior education services should be drawn more clearly, to be more acceptable for readers.

8. Section 3.1 Analysis could be expanded with more details to clarify the analysis method.

9. You emphasize interaction objectives. "As the basis for the evaluation of Interaction effectiveness, the analysis perspective of Interaction objectives needs to start from the four levels of home care service needs."  Please state which four levels.

10. The name of the subsection "Practical analysis of the Interaction model" could be changed to Discussion.

11. The Conclusion section should be adjusted so that it is clear what your scientific contribution is.

12. Be sure to check for typographical and linguistic errors.

Round 2

Reviewer 2 Report

It would be good if Subsection 3.2 Design and development changed the title to Design.